# Warm-Up Strategies and Effects on Performance in Racing Horses and Sport Horses Competing in Olympic Disciplines

**DOI:** 10.3390/ani14060945

**Published:** 2024-03-19

**Authors:** Thibault Frippiat, Dominique-Marie Votion

**Affiliations:** 1Fundamental and Applied Research for Animals & Health (FARAH), Faculty of Veterinary Medicine, University of Liège, 4000 Liège, Belgium; dominique.votion@uliege.be; 2Sportpaardenarts—Equine Sports Medicine, 1250AD Laren, The Netherlands

**Keywords:** horses, exercise, warm-up exercise, sports, athletic performance, animal physical conditioning, physiological adaptations, veterinary sports medicine, animal welfare

## Abstract

**Simple Summary:**

Warm-up, a routine part of the physical preparation for exercise and competition, ensures the adaptation of body systems from rest to exercise with the dual aim of improving performance and reducing the risk of injury. Passive warm-up techniques (by external means) are not often implemented and very little studied in equestrianism. This scoping review aimed to summarize active warm-up strategies (by a gradual increase in exercise intensity) and effects on equine performance from peer-reviewed publications from 1996 to January 2024. An adequate warm-up generated, among others, an increase in body temperature and cardiorespiratory adaptations to exercise, such as higher heart rates, faster oxygen consumption by muscles, and less blood and muscle lactate accumulation. A low-intensity warm-up regimen induced identical beneficial effects as a high-intensity regimen. Different warm-up strategies were observed between dressage and show jumping horses, while few studies described warm-up strategies in eventing and racing horses. Dressage horses were warmed up longer than show jumping horses. Warm-up duration and intensity increased with an increasing competitive level in dressage and show jumping horses, without affecting the final score. In conclusion, this review emphasizes the low level of current evidence on the best warm-up strategies per equestrian discipline and level.

**Abstract:**

Warm-up is a standard component of exercise preparation, intended to lower the risk of injury and improve performance. Comprehensive evidence-based guidelines per discipline are missing. This scoping review aimed to describe the physiological effects and strategies of active warm-up in horses according to different equestrian disciplines. The search strategies identified 479 papers for review. After application of selection criteria, 23 articles published from 1996 to January 2024 were included of which 12 discussed the effects of warm-up on physiological parameters and 11 discussed warm-up strategies in different disciplines. As shown in humans, warm-up enhanced aerobic capacity and increased blood and muscle temperatures, independently from its intensity. Riders emphasized the importance of warm-up to prepare horses for physical work and to increase their reactiveness to aids. A canter or trot was the preferred gait in elite or non-elite dressage horses, respectively, while the walk was in show jumping horses. Warm-up duration and intensity increased with increasing competitive level, but a longer and/or more intensive warm-up did not affect the final score. Dressage riders warmed up their horses for a longer time compared to show jumping riders. Future studies should objectively establish the most profitable warm-up strategies per equestrian discipline and level.

## 1. Introduction

Irrespective of the equestrian discipline, the main objective of conditioning horses consists in improving performance while preserving their health and well-being. In this context, the concept of equine welfare has evolved into a principle that all stakeholders engaged in equine performance, including trainers, riders, groomers/caretakers, judges, and stewards, should adopt [1,2,3,4,5]. To improve performance, various factors come into play, with training occupying an important position. The warm-up phase is part of the preparation for exercise, with the expected goals of reducing the risk of injury during exercise and enhancing performance through a gradual transition from rest to exercise [6,7]. Different physiological adaptations to exercise occurring during warm-up have been observed in humans, supporting its positive effect on subsequent performance [8,9,10,11,12].

One of the main outcomes is an increase in muscle temperature as a result of friction within the sliding filaments during muscular contraction, the metabolism of muscle fuels, and the dilatation of intramuscular blood vessels [6,13,14]. As muscle temperature increases, several responses are initiated within the body such as enhanced muscle metabolism, increased blood circulation to working muscles (Figure 1) resulting in an enhanced oxygen supply, and increased capacity of working muscles to extract and use oxygen [15,16]. Elevating tissue temperature results also in faster nerve conduction, improving the rate and reaction time of muscle contraction [17] and an increase in the elasticity of muscles, tendons, and ligaments, which may reduce the risk of injury and allow for a full range of motion in the joints [18,19,20]. In humans, antagonist muscles are the most frequently torn muscles during activity that has not been preceded by a warm-up, as they relax slowly and incompletely when agonist muscles contract [15].

The second main outcome of warm-up is enhanced aerobic metabolism. During warm-up, epinephrine and norepinephrine are released [22]. As a consequence, tissue oxygenation is improved by an increase in heart rate (HR) (Figure 2), breathing frequency, and tidal volume and by splenic contraction and the subsequent release of stored red blood cells into the circulation [23,24,25,26,27]. The energy supply to muscles is enhanced by the activation of glycogenolysis and lipolysis [28].

In humans, studies on warm-up strategies have focused on physical preparation for specific sports or sports categories [29]. Human athletes perform in highly different disciplines, e.g., from very short strength exercises to longer endurance sports, from individual to team sports, and in different environments. Most equine studies on warm-up focused on racing horses and sport horses competing in Olympic disciplines (dressage, show jumping, and eventing), which require strength and stamina, and rely mainly on aerobic metabolism. Human preparation to exercise involves raising muscle or core temperature by both passive (using some external means) and active (using exercises) warm-ups [30,31].

Understanding the effects of warm-up in horses is key to unlocking their true exercise potential and preventing injuries, thus contributing to their welfare. Common warm-up practices rely mainly on traditions, while comprehensive evidence-based guidelines per equestrian discipline and level are scarce. This paper aims to review the literature to describe (1) the physiological effects of active warm-up in horses and (2) the discipline-specific active warm-up strategies in racing horses (Standardbreds and Thoroughbreds) and sport horses competing in Olympic disciplines (dressage, show jumping, and eventing).

**Figure 2 animals-14-00945-f002:**
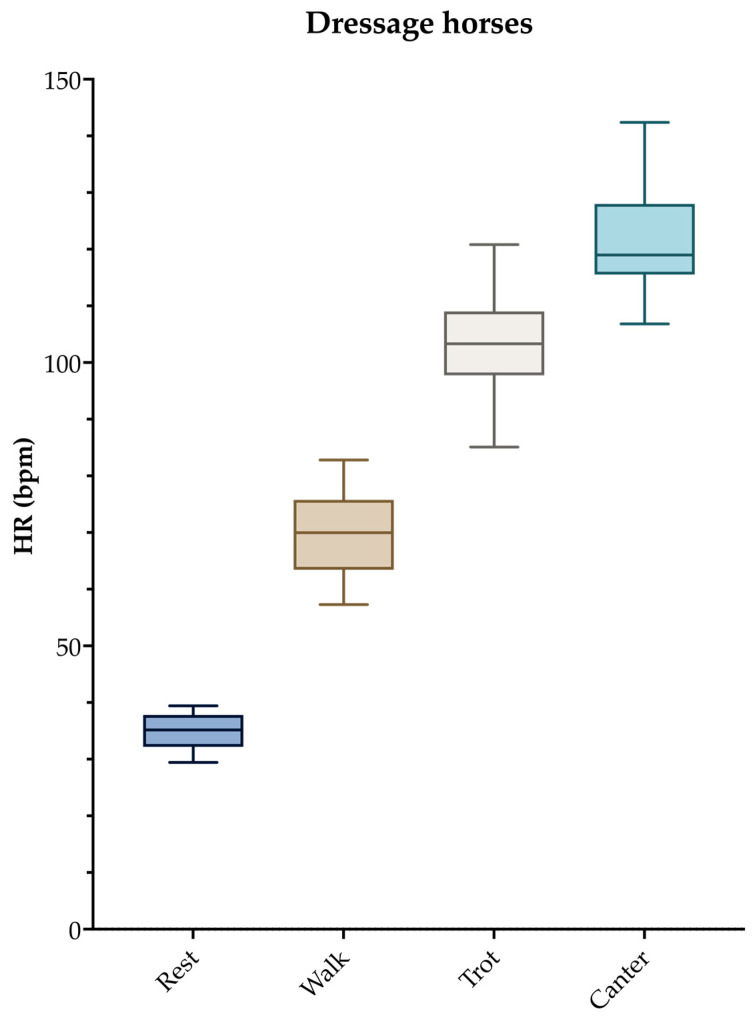
Higher heart rates (HR) are observed with increasing exercise intensity in dressage horses during warm-up at different gaits. Adapted from [32] with permission from Brill.

## 2. Materials and Methods

A systematic search of the literature on warm-up in horses published between 1983 and January 2024 was performed. To identify all published papers, two search methods, “traditional method” and “search engine method”, were used. Both methods were conducted by one investigator (T.F.) using the following keywords and Boolean operators: [“horse” OR “horses” OR “pony” OR “ponies” OR “equine”] AND [“warm up” OR “warm-up” OR “warming up” OR “warming-up”]. The searches were performed on 10 June 2023 and 12 February 2024.

The traditional method was conducted using a general search function with the keywords in Google Scholar and the University of Liège library website. Publication titles were scanned for relevance to warm-up in horses or ponies, and a list of relevant articles was created in a Microsoft Excel file. The search engine method was conducted using the same keywords as the traditional method to search for all studies in three different databases (CAB Direct, PubMed, and Scopus). The advanced search was used in these databases, where the above keywords were used within “article title, abstract, and keywords” only. The titles of articles were also exported into the Microsoft Excel file. Then, the titles from both search methods were combined to create a new dataset, and duplicates were removed. The hyperlinks to the accessed websites are shown in Appendix A.

The last step of the systematic search was article selection. Both investigators independently read the title and abstract of the selected articles and, based on the selection criteria (Table 1), decided whether to retain the article for further analysis. Articles accepted by the two investigators were automatically retained for further analysis, while those accepted by only one investigator were first discussed by the two investigators before being retained for further analysis. Two categories of articles were created, considering their study fields, aims, and methodologies: (1) articles discussing the physiological effects of warm-up in horses and (2) articles discussing warm-up strategies within the scope of included equestrian disciplines. Following the categorization of articles, all selected articles were inserted in the artificial intelligence tool ResearchRabbit to search for additional publications that remained unnoticed by the traditional and search engine methods.

## 3. Results

### 3.1. General Review Statistics

A total of 479 records were identified through all search methods (Figure 3). Search with ResearchRabbit did not provide any other relevant publication for the review that was not yet included. Following the removal of duplicate records, 243 records were screened for relevance to the review. After title and abstract screening, a total of 196 publications were evaluated in full. A large proportion of studies were excluded as they were not investigating warm-up or because the warm-up strategy was incompletely or not objectively described.

After completion of the selection process, 23 publications were included, among which 12 discussed the effects of warm-up regimens on physiological parameters such as core temperature, aerobic metabolism, and/or acid-base balance (Table 2), and 11 discussed warm-up strategies in different equestrian disciplines based mainly on questionnaires and observations at shows (Table 3). Of the 12 records investigating the effects of warm-up on physiological parameters, 10 (83%) were randomized controlled crossover trials, and 2 (17%) were prospective observational studies. Of the 11 records investigating warm-up strategies, 1 (9%) was a randomized controlled crossover trial, 1 (9%) was based on a questionnaire, and 9 (82%) were observational studies.

### 3.2. Effects of Warm-Up on Aerobic Metabolism

#### 3.2.1. In Standardbreds

Three studies focused on the effects of warm-up regimens on aerobic metabolism in Standardbreds [35,36,37]. Different parameters were used: VO_2_ (oxygen consumption or aerobic capacity), VCO_2_ (rate of elimination of carbon dioxide), time to fatigue, and blood lactate accumulation. A 5 min warm-up at 50% of VO_2max_ (mean 4.4 m/s) resulted in an acceleration of the kinetics of gas exchange in 13 Standardbreds [35]. In this crossover setting, the calculated relative proportions of the total energy supplied by aerobic and anaerobic sources were different with (80% and 20%, respectively) or without prior warm-up (73% and 27%, respectively). The VO_2_ and VCO_2_ kinetics were faster in horses having performed a warm-up before the exercise test at 115% of VO_2max_ on a treadmill. However, the time to fatigue was not different with versus without prior warm-up, and blood lactate accumulation was higher in horses having performed a warm-up.

In two studies each involving six Standardbreds, the effects of a low- (10 min at 50% of VO_2max_) and a high-intensity warm-up (5 or 7 min at 50% of VO_2max_, followed by 45 s intervals at 80%, 90%, and 100% of VO_2max_) confirmed the faster kinetics of VO_2_ and VCO_2_ after warm-up [36,37]. Both warm-up regimens lowered the accumulated oxygen deficit and the rate of blood and muscle lactate accumulation during exercise compared to the situation without prior warm-up [36,37]. The warm-up was associated with higher aerobic energy contribution to the total energy expenditure, lower glycogenolysis, and longer run time to fatigue [36]. The increase in run time to fatigue was the highest following a low-intensity warm-up (47%), but it was still higher after a high-intensity warm-up (30%) compared to exercise without prior warm-up [36].

#### 3.2.2. In Thoroughbreds

Three studies focused on the effects of different warm-up regimens on aerobic metabolism in Thoroughbreds [34,38,39]. The used parameters were mainly VO_2_, VCO_2_, time to fatigue, blood lactate, and HR. In a crossover design, Mukai et al. [38] compared a moderate- (1 min at 70% of VO_2max_) to a high-intensity warm-up (1 min at 115% of VO_2max_) before a sprint until fatigue at 115% of VO_2max_ in 11 Thoroughbreds. Both warm-up regimens induced a higher VO_2_ during the sprint and a lower blood lactate accumulation during the first minute of exercise compared to the situation without prior warm-up. In another study, the same team compared the effects of other warm-up intensities and showed that a high-intensity warm-up (120 s at 100% of VO_2max_) accelerated VO_2_ kinetics and reduced reliance on net anaerobic power at the onset of the subsequent sprint compared to low- (400 s at 30% of VO_2max_) or moderate-intensity warm-up (200 s at 60% of VO_2max_) in nine Thoroughbreds [39]. Lund et al. [34] showed that a low-intensity warm-up was sufficient to provide a beneficial effect on VO_2_ in six Thoroughbreds. Both the low- (5 min walk, 400 m canter, 5 min walk) and high-intensity warm-up (5 min trot, canter until venous temperature > 39.5 °C, 5 min trot) resulted in a decrease of about 3% in VO_2_ during exercise to fatigue compared to previous measurements without a prior warm-up.

#### 3.2.3. In Sport Horses

No studies on aerobic metabolism were found in dressage, show jumping, or eventing horses. One study regarding six Mangalarga Marchador horses showed no effect of a 10 min walking warm-up on parameters of aerobic metabolism (HR, breathing frequency, and blood lactate), measured directly after a 50 min Marcha test, a predominantly aerobic and physiologically stressful exercise of moderate intensity (12 km/h) [40]. However, HR recovered faster after exercise when a prior warm-up had been performed.

### 3.3. Effects of Warm-Up on Thermoregulation

#### 3.3.1. In Standardbreds

Two studies focused on the effects of warm-up regimens on thermoregulation in Standardbreds [36,37]. Warm-up exercise was associated with an increase in both muscle and blood temperatures [36]. Middle gluteal muscle temperature increased with averages of 1.7 °C and 3.4 °C after, respectively, a low- (10 min at 50% of VO_2max_) and a high-intensity warm-up (5 or 7 min at 50% of VO_2max_ followed by 45 s intervals at 80%, 90%, and 100% of VO_2max_) [36]. Right atrial blood temperature showed parallel but slighter increases of, respectively, 0.9 °C and 1.6–1.9 °C [36,37]. These increases were maintained throughout the high-intensity exercise at 115% of VO_2max_ on a treadmill [37].

#### 3.3.2. In Thoroughbreds

Three studies focused on the effects of warm-up regimens on thermoregulation in Thoroughbreds [34,38,39]. Mukai et al. [38,39] showed that all warm-up intensities (i.e., low, moderate, and high) induced an increase in blood temperature measured in the pulmonary artery. This increase was maintained throughout the whole sprint at 115% of VO_2max_ on a treadmill. However, a high-intensity warm-up induced a higher increase in blood temperature than low- or moderate-intensity warm-up regimens [39].

Without, or after a light (5 min walk, 400 m canter, 5 min walk) or heavy warm-up (5 min trot, canter until venous temperature > 39.5 °C, 5 min trot), approximately 12.8, 15.1, and 18.4 MJ of heat, respectively, were generated in response to warm-up and exercise at 105% of VO_2max_ in six Thoroughbreds [34]. The low-intensity warm-up had beneficial effects on heat balance, including a slower accumulation of heat, despite a higher body temperature at the onset of maximal exercise. Furthermore, sweating was initiated earlier during low-intensity warm-up, which promoted better thermoregulation.

#### 3.3.3. In Sport Horses

Two studies focused on the effects of warm-up on thermoregulation in sport horses competing in Olympic disciplines [41,42] and one study on Mangalarga Marchador horses [40]. Buchner et al. [41] showed no effect for a 10 min whole-body vibration warm-up or two exercising warm-up regimens (10 min walk, or 8 min walk, and 1 min trot, respectively) on core temperatures in 10 horses. The exercise warm-up regimens induced a slight increase in skin temperature, while the whole-body vibration protocol did not. Heart rate was not modified after any of the warm-up regimens, while a slight increase in breathing frequency was observed after the exercise warm-up.

In another study, the effects of four warm-up regimens of increasing duration (10 min walk, 5, 10, 15, or 20 min trot, and 5 min walk) on body and surface temperatures were observed in 12 Warmblood horses (6 leisure horses and 6 jumping sport horses) [42]. In all horses, the rectal temperature increased after each type of warm-up and was higher after the longest warm-up compared to other warm-up durations. Superficial temperatures were acquired by thermography and increased with increasing warm-up duration. Palmar and plantar surfaces of distal limb parts were warmer than dorsal surfaces, with forelimbs being warmer than hind limbs. The increase in surface temperature in distal limb parts was greater in jumping sport horses than in leisure horses.

A 10 min walking warm-up showed no effect on the rectal temperature measured directly after a 50 min Marcha test in six Mangalarga Marchador horses [40].

### 3.4. Effects of Warm-Up on Acid-Base Balance and Biochemistry

Three studies reported the effects of warm-up on biochemistry parameters and acid-base balance in horses [43,44,45]. In a crossover design, Frey et al. [43] tested the effects of the administration of sodium bicarbonate on blood pH, base excess, bicarbonate, and electrolytes during two types of warm-up (2-mile slow or 1-mile fast) and racing in 12 Standardbreds. Following the slow warm-up, venous blood acid-base balance, sodium or chloride did not change, while potassium increased and calcium decreased. Following the fast warm-up, bicarbonate, base excess, pH, and calcium decreased, and potassium increased, while sodium and chloride remained unchanged. The administration of sodium bicarbonate increased venous blood bicarbonate, base excess, pH, sodium, chloride, potassium, and calcium during warm-up, racing, and recovery.

In two studies on 10 and 7 healthy Italian saddle horses, the effects of a 15 min warm-up (pacing, trotting, galloping, and six jumps at 1.00–1.40 m) on biochemistry parameters were analyzed [44,45]. While base excess was increased, no difference in bicarbonate or pH was observed after warm-up [44]. Furthermore, warm-up induced a decrease in plasma glucose concentration and an increase in serum aspartate aminotransferase, alanine aminotransferase, creatine kinase, creatinine, and potassium concentrations [45]. No changes were observed in alkaline phosphatase, gamma-glutamyltransferase, lactate dehydrogenase, urea, total bilirubin, sodium, or chloride after warm-up in these horses [45].

A 10 min walking warm-up showed no effect on serum aspartate aminotransferase, creatine kinase, cortisol, or plasma glucose measured directly after a 50 min Marcha test in six Mangalarga Marchador horses [40].

### 3.5. Warm-Up Strategies

#### 3.5.1. In Racing Horses

One field study described warm-up strategies and their effects on performance in four Standardbreds and three Thoroughbreds [46]. Short and long warm-up regimens were compared. In Standardbreds, an increase in HR and breathing frequency was observed directly after warm-up and 15 min after exercise, following the long warm-up, but not after the short warm-up. Rectal temperature and body weight loss were increased after both warm-up regimens but were higher after the long warm-up. In Thoroughbreds, there was an increase in HR, breathing frequency, rectal temperature, and body weight loss after both warm-up regimens, without a significant difference between warm-up regimens.

#### 3.5.2. In Dressage Horses

Five studies described warm-up strategies in dressage horses and their effects on performance [47,48,49,50,51]. The mean warm-up durations in these studies are presented in Table 4. A description of dressage levels is shown in Appendix B.

Williams et al. [48] observed HR and warm-up duration for 36 Elementary and 14 Medium British Dressage tests in 35 horses. For different Elementary tests, the warm-up duration was different between horses (range: 18 ± 7 to 53 ± 17 min; mean: 31.3 ± 15.4 min), but no difference in the mean (91 ± 13 bpm) or peak HR (146 ± 35 bpm) during warm-up was observed. Horses performing at the Elementary level spent 35.5% of the time at an HR of 80–120 bpm, 29.7% at an HR of 120–160 bpm, and 1.3% at an HR above 160 bpm. For different Medium tests, there was no difference in the warm-up duration between horses (mean: 31.4 ± 10.0 min) and in the mean (91 ± 10 bpm) or peak HR (144 ± 32 bpm) during warm-up. Horses performing at the Medium level spent 38.1% of the time at an HR of 80–120 bpm, 31.6% at an HR of 120–160 bpm, and 0% at an HR above 160 bpm. A positive correlation was observed between the mean HR during warm-up and competition at both levels.

In a study comparing warm-up patterns of 12 elite (Intermediate I level and above) versus 20 non-elite (Medium level and below) dressage horses, the mean warm-up duration (elite: 15.7 ± 5.8 min; non-elite: 15.1 ± 6.4 min) and time in walk (elite: 4.4 ± 2.5 min; non-elite: 5.2 ± 4.5 min) did not differ between groups [50]. Elite horses spent more time in canter (4.6 ± 2.2 min) than non-elite horses (2.5 ± 1.3 min). Non-elite horses spent more time in trot (7.3 ± 3.4 min) than elite horses (6.3 ± 3.0 min). The main gait during warm-up was trot in both elite and non-elite horses. No difference in the time spent on left and right reins at walk, trot, and canter was observed in the two groups. Another study involving seven riders performing 39 warm-up sessions at home showed riders spent the most time in a walk (10 ± 4 min) and a trot (7 ± 4 min) and the least time in a canter (4 ± 3 min) [51]. Most riders walked their horses with a low-head carriage during the first walk phase. No difference was found in the total warm-up duration or the total time spent in a walk between warm-up sessions performed in air temperatures below 5 °C and above 40 °C.

Another study focused on the association of warm-up patterns with the level and final score [47]. The warm-up duration increased with increasing competitive level (Novice: 24.4 ± 10.0 min; Medium: 31.5 ± 11.5 min; Prix St-Georges: 32.9 ± 11.3 min; Grand Prix: 34.6 ± 10.2 min). A trot was the main gait during the warm-up of Novice competitors and a walk for all other competitors. Prix St-Georges (30%) and Grand Prix (28%) competitors spent more time at a canter than Novice (20%) and Medium (24%) competitors. No effect of rider experience was detected on the warm-up strategy. A positive association between the total warm-up time and the final score was observed for Novice and Prix St-Georges competitors but not for Medium and Grand Prix competitors.

In an online survey, 139 European dressage riders were asked about their decision-making when warming up a horse at home and before a competition [49]. The main reported reasons for performing a warm-up were to prepare the horse’s musculoskeletal system for physical work, to increase the horse’s reactiveness to the rider’s aids, and to increase suppleness. Only 23% of riders used a fixed warm-up regimen at home. Most dressage riders (65%) reported a walk as the main gait during the warm-up at home, while 54% believed it was beneficial to use the same warm-up routine at home and before a competition. According to most dressage riders, a warm-up should last between 10 and 20 min in length.

#### 3.5.3. In Show Jumping Horses

Six studies described warm-up strategies in show jumping horses and their effects on performance [49,51,52,53,54,55]. The mean warm-up durations in these studies are presented in Table 4. Whitaker et al. [52] showed that warm-up was shorter in 49 Novice (0.90 m competition; 15.0 ± 0.5 min) compared to 38 Intermediate competitors (1.20 m competition; 17.8 ± 0.6 min). Novice competitors spent less time walking (2.5 ± 0.3 min) than Intermediate competitors (3.8 ± 0.3 min). Furthermore, the total jumped fences and total successfully jumped fences during warm-up were lower in Novice (total jumped: 9.6 ± 0.3; total successful: 8.5 ± 0.3) compared to Intermediate competitors (total jumped: 13.1 ± 0.4; total successful: 11.8 ± 0.3). In a study investigating 45 warm-up sessions for a 1.30 m show jumping competition across 27 riders and 20 horses, the warm-up duration varied greatly amongst riders (range 4–63 min; mean 18.7 ± 12.4 min) [55]. A walk was the most common gait, and a trot was the least prevalent gait used during warm-up. From 2 to 15 jumps were used during warm-up, and the performance in the show ring, expressed as the number of faults, was not associated with the warm-up routine. However, another study of 82 competitors found that, while warm-up duration did not affect the score in the ring, more jumps and higher obstacles during warm-up decreased performance [54]. Tranquille et al. [53] found no difference in the warm-up duration, the time in each pace, and on each rein, mean, peak, and final warm-up HR in 10 elite horses over three consecutive days. Most horses spent more time in a left canter, which was the preferred lead in 50% of horses when landing and leaving the fence. Horses cantered slower, with a shorter stride length and longer stride duration during warm-up compared to the course. Another study involving three riders performing 22 warm-up sessions at home showed riders spent the most time in a walk (15 ± 7 min) and a trot (8 ± 2 min) and the least time in a canter (4 ± 2 min) [51]. Most riders walked their horses with a low-head carriage during the first walk phase. No difference was found in the total warm-up duration or the total time spent in a walk between warm-up sessions performed in air temperatures below 5 °C and above 40 °C.

In an online survey, 125 European show jumping riders were asked about their decision-making when warming up a horse at home and before competition [49]. The main reported reasons for performing a warm-up were to prepare the horse’s musculoskeletal system for physical work, to increase the horse’s reactiveness to the rider’s aids, and to decrease the risk of injury. Only 30% of riders used a fixed warm-up regimen at home. A trot was reported by 46% of show jumping riders as the main gait during the warm-up at home, while 46% believed it was beneficial to use the same warm-up routine at home and before a show. According to show jumping riders, a warm-up should last between 10 and 20 min in length. Before the show, 49% of riders used 7–10 fences to warm up, 41% used 4–7 fences, 9% used less than 4 fences, and 1% did not jump fences before entering the show ring.

#### 3.5.4. In Eventing Horses

One study described warm-up strategies in 10 eventing horses at a two-day eventing competition [56]. Mean warm-up durations for all three parts of the competition in Intermediate and Advanced levels are presented in Table 4. No difference was found in the warm-up duration between levels for dressage and cross-country tests, while horses performing at the Intermediate level warmed up for significantly shorter times (16 ± 9 min) than horses performing at the Advanced level (32 ± 14 min) for show jumping tests. No difference was observed between competition levels in the mean or peak HR during the warm-up of all three tests.

## 4. Discussion

Warm-up techniques can be qualified as either passive or active. Passive warm-up relies on external factors to elevate muscle temperature and prime the body for subsequent physical activity without engaging in active muscular work that consumes energy substrate stores as is the case with active warm-up [29,31].

Passive warm-up often includes external heat application, massage, or exposure to environmental conditions that promote thermal elevation. The rationale behind these approaches lies in their ability to increase blood flow, leading to improved oxygen delivery to muscles and enhanced metabolic reactions. Additionally, passive warm-up techniques may influence the viscoelastic properties of muscles and tendons, potentially reducing stiffness and increasing joint range of motion during subsequent exercise [20,57,58,59]. However, passive warm-up does not always result in increased muscle temperature in humans [7,60]. Furthermore, increasing superficial temperature and dilating cutaneous blood vessels could divert a large amount of blood to the skin rather than to working muscles [7]. While it improves short-duration (<10 s) dynamic force [61], passive warm-up does not seem to improve isometric force in humans [62] and might even be detrimental to long-term performance (>5 min) [63,64]. Of the 23 studies included in the present review, only one investigated the effect of passive warm-up in horses [40]. Questionnaires among riders and trainers of racing and sport horses do not identify passive warm-up as a common feature within the preparation of horses for exercise or competition [46,49].

In humans, the effectiveness of an active warm-up strategy is determined largely by its composition (i.e., intensity and duration) as well as the length of the transition phase [29]. In the Results section of this scoping review, we described the effects of active warm-up on physiological parameters (aerobic metabolism, thermoregulation, and acid-base balance and biochemistry) and the warm-up strategies in racing and sport horses competing in dressage, show jumping, and eventing. Hereunder, we discuss the effects of warm-up on equine performance and the evidence for the optimization of warm-up routines for equestrian disciplines.

### 4.1. Effects of Warm-Up on Performance

#### 4.1.1. By Means of Thermoregulation

A higher muscle temperature could contribute to enhanced aerobic capacity for energy production, through an increase in muscle VO_2_ from a faster metabolic rate-limiting the muscular reactions (Q_10_ effect) of oxidative phosphorylation, and an enhanced oxyhemoglobin dissociation increasing oxygen availability for the muscle [65,66]. A 1 °C increase in muscle temperature enhanced subsequent exercise performance by 2–5% in humans [61,67], through the increase in ATP turnover, the muscle cross-bridge cycling rate, and oxygen uptake kinetics, resulting in enhanced muscular function [16,29].

Of eight studies on equine thermoregulation included in the present review, seven demonstrated an increase in core temperature (i.e., blood or muscle temperature) after different regimens of warm-up in racing and sport horses [34,36,37,38,39,40,42]. One study showed no effect of warm-up on core temperature [41], but this study involved relatively low intensities of warm-up regimens, which could explain the conflicting finding.

The increase in muscle temperature after a warm-up enhanced the aerobic energy contribution during sprints in racing horses, resulting in a longer run time to fatigue compared to exercise without a prior warm-up [36]. The increase in blood temperature was subsequently maintained throughout the whole sprint [38,39]. In exercising Thoroughbreds, a warm-up activated the thermoregulation mechanisms, observed by an enhanced onset of sweating [34]. Despite a higher mean body temperature at the onset of maximal exercise following a warm-up, the subsequent accumulation of heat was not as rapid.

#### 4.1.2. By Means of Enhanced Aerobic Metabolism

Elevating body temperature is not the sole determinant of energy metabolism changes during exercise. Changes in the mechanisms underlying both aerobic and anaerobic metabolism contribute to improved subsequent exercise performance [68,69,70]. The rate of increase in VO_2_ during high-intensity exercise is much greater in horses than in humans [71]. All six studies on gas exchange included in the present review showed a beneficial effect of warm-up on VO_2_ kinetics and/or aerobic energy contribution [34,35,36,37,38,39]. This change in oxygen uptake kinetics enhances the ability of muscles to work aerobically and reduces blood lactate accumulation during high-intensity exercise [36,37,38].

Six of eight studies investigating changes in blood lactate included in the present review showed increased lactatemia immediately after warm-up and lower lactatemia during subsequent exercise and recovery [34,36,38,39,44,45]. Conversely, one study found no change in lactatemia following a warm-up compared to exercise without a prior warm-up [40]. The conflicting finding of this latter study could be attributable to the discipline and required exercise involved (i.e., a 50 min Marcha test). In one study, however, blood lactate accumulation was higher in horses having performed a prior warm-up [35], probably due to an accumulation of blood lactate during the warm-up to produce a significant difference between the two groups after exercise. Peak blood lactate concentrations occur between 1 and 10 min after exercise [72]; thus, the horses that did not perform a prior warm-up may have peaked later than the post-exercise sampling.

### 4.2. Optimization of Warm-Up

#### 4.2.1. Intensity

In humans, a high-intensity warm-up did not enhance sprint cycling performance compared to a low-intensity warm-up followed by a few sprints [73,74]. Contradictorily, a more intensive warm-up produced better performance than a low-intensity warm-up in soccer players [75]. Metabolic acidemia following a warm-up of too high intensity has been shown to impair supramaximal performance and reduce the accumulated oxygen deficit [76].

Warm-up increased the time to fatigue in racing horses [35,36,38]. However, a high-intensity warm-up did not seem to provide an additional advantage for subsequent sprints compared to a low-intensity warm-up [35,38]. In one study, the run time to fatigue after high-intensity warm-up was not higher than without warm-up, suggesting it may be critical to structure the warm-up to enhance VO_2_ kinetics without amplifying factors that lead to fatigue [36].

In show jumping horses, although increased warm-up duration was recommended with the increased complexity of competitive exercise [53,54], an intensive warm-up decreased performance [54]. A higher number of jumps during the warm-up was associated with more penalty points and a lower ranking during competition [54].

#### 4.2.2. Duration

In humans, a longer warm-up did not enhance sprint cycling performance compared to a short warm-up followed by a few sprints [73,74]. The warm-up duration also did not influence subsequent exercise in handball and rugby players or during an anaerobic exercise test [77,78,79]. However, a shorter warm-up produced better performance than a longer warm-up in soccer players [75]. These findings emphasize the different physical preparations needed for different sports, although a longer warm-up could deplete energy stores and decrease heat storage capacity [80]. While a longer warm-up had a deleterious effect on perceived exertion and the test duration [81,82], reducing its duration resulted in higher peak power outputs during an anaerobic test, likely due to reduced fatigue [83].

In show jumping horses, the combined mean and standard deviation (SD) of warm-up duration was 20.5 ± 8.6 min when merging the results of five studies amongst different levels included in the review [51,52,53,54,55]. While warm-up duration varied greatly amongst riders [49,53,55], it did not seem to affect the final score [54]. Non-elite horses warmed up longer than elite horses [52].

Compared to show jumping horses, dressage horses warmed up longer, with a combined mean and SD of 28.1 ± 11.8 min when merging the results of four studies amongst different levels [47,48,50,51]. Contrariwise to show jumping, dressage riders at higher competitive levels warmed up longer compared to riders at lower levels [47,50]. The warm-up duration was associated with higher final scores at different levels. Still, too great of a warm-up intensity and/or duration could, however, result in early fatigue and reduced performance [47,50].

A long warm-up regimen was more advantageous in terms of increased HR and temperature than a short warm-up regimen in Standardbreds, while it was not in Thoroughbreds [46].

In eventing horses, warm-up strategies vary depending on the specific test being performed [56]. Although no significant effect of the level of competition was observed on the warm-up strategy for dressage and cross-country tests, it was found that horses competing at higher levels were warmed up longer for show jumping tests compared to those at lower levels, in contradiction to the findings from a previous study conducted on show jumping horses [52].

#### 4.2.3. Composition

In humans, specific warm-up strategies in terms of the number of repetitions, training load, and/or composition of warm-up influenced subsequent performance [84,85,86,87]. While several studies described the time spent in each gate and rein, stride characteristics, and/or total practice fences jumped [47,50,52,53], no study determined the effects of the specific intensity and type of exercise in different equestrian disciplines and levels.

Additionally, it is questionable whether riders can correctly assess the composition of warm-up, as only 13.6% of riders accurately recalled the routines of their warm-up at home [51]. Furthermore, when questioned, show jumping riders reported a trot as the main gait during warm-up [49]. Still, observational studies described conflicting findings about the main gait during warm-up, showing the lack of homogeneity in warm-up practices [51,52,53,55]. When cantering, horses spent more time at the left rein [53].

In contrast to show jumping riders, dressage riders reported the walk as the primary gait during warm-up [49]. Two observational studies showed that the walk was indeed the predominant gait during warm-up in all levels except the Novice level [47,51], while a trot was in both elite and non-elite dressage horses in another observational study [50]. An increased proportion of the canter was generally observed during the warm-up of the elite compared to the non-elite dressage horses, which performed more advanced movements than non-elite horses [47,50]. The horse’s level of training could also influence the effects of a warm-up, as the effect of warm-up on temperature was achieved earlier and lasted longer in heavily trained horses compared to non-performance horses [42].

#### 4.2.4. Transition Phase

No studies included in the present review investigated the transition phase, defined as the period between the warm-up and the competition, and its effect on subsequent exercise. It is not uncommon that horses have to wait for longer periods than expected or that horses have to wait between two rounds with an unknown interval duration. In humans, both too-long and too-short transition phases have been related to the impairment of subsequent performance [70,88]. Reducing the transition duration from 45 to 10 min was associated with improvements (of about 1.4%) in 200 m swimming performance in humans [89,90]. Muscle temperature declines immediately after exercise cessation, with appreciable declines occurring as early as 15 min post-exercise in humans [90,91]. During lengthy transition phases, passive heat maintenance techniques could preserve the beneficial temperature effects induced by a prior active warm-up.

#### 4.2.5. Warm-Up at Home versus at a Show

As exercise intensity, duration, and/or composition are possibly different for training at home and during a competition, the optimal warm-up regimen for the preparation of both physical activities could differ between both situations. To the authors’ knowledge, no study has focused on this feature in humans or animals. In a survey, about half of dressage and show jumping riders believed the same warm-up regimen at home or a show could be beneficial [49]. Less than a third of the respondents used a fixed warm-up regimen when training at home. Riders reported adapting the warm-up practice to the temperament and age of the horse and the time of the year and day. However, in another study, air temperature did not influence riders’ warm-up strategy [51].

### 4.3. Limitations

Several studies included in the current review present limitations, such as a low number of included horses, missing data, and/or possible bias due to data acquisition by riders. Two studies in the review used questionnaires, with possible survey bias. The review itself presents limitations; e.g., studies are not directly comparable as they used different parameters, methods, and data analysis.

Equestrianism is a sport involving a horse–rider combination. The present review only focused on the horse’s warm-up. However, the rider’s physical and mental preparation could also play a role in the performance of the horse–rider combination.

Notwithstanding that warm-up and cool-down are frequently associated in the description of optimal training routines in humans, there are conflicting data about the usefulness and best practices in humans [92,93,94]. As little evidence is available on cool-down in horses, the present review did not include cool-down practices.

## 5. Conclusions

A warm-up induced faster kinetics of VO_2_ and VCO_2_, less blood and muscle lactate accumulation, increased blood and muscle temperatures, and a higher plasmatic potassium concentration. These changes were observed with all intensities of warm-up, but a low-intensity warm-up was sufficient to induce these beneficial effects. Both dressage and show jumping riders reported warm-up as important to prepare the horse’s musculoskeletal system for physical work and to increase the horse’s reactiveness to the rider’s aids. Observational studies showed differences in warm-up strategies depending on the discipline and level. The walk is the most common gait in show jumping horses, but a canter and a trot are in elite and non-elite dressage horses, respectively. While the warm-up duration and intensity increased with increasing competition level, they did not seem to affect the final score.

This scoping review on the effects and strategies of warm-up highlights the paucity of information on horses. Future studies must objectively establish the most profitable warm-up strategies in the different equestrian disciplines and levels, including the intensity and duration of warm-up, the effects of practicing specific movements, and the possible consequences of a transition phase between warm-up and competition.

## Figures and Tables

**Figure 1 animals-14-00945-f001:**
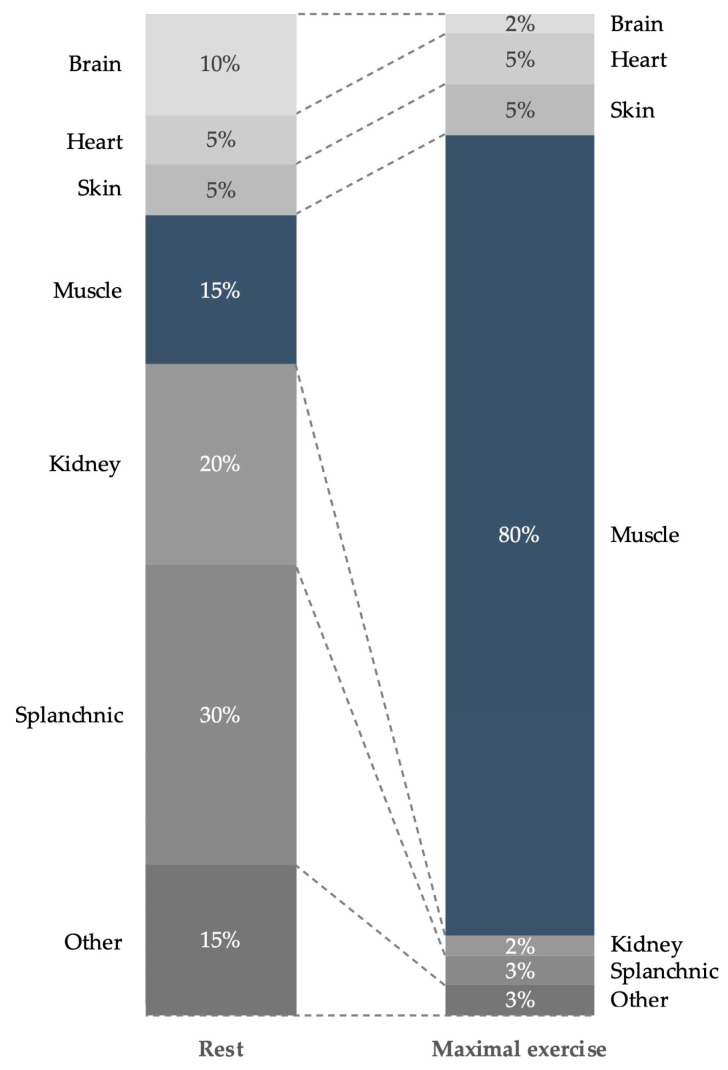
During warm-up, physiological changes are initiated such as the distribution of cardiac output (blood flow) to skeletal muscles and other organs as illustrated in rest (**left**) and maximal exercise (**right**). Note the largely increased cardiac output toward skeletal muscles (dark blue). Adapted from [21] with permission from John Wiley & Sons, Inc.

**Figure 3 animals-14-00945-f003:**
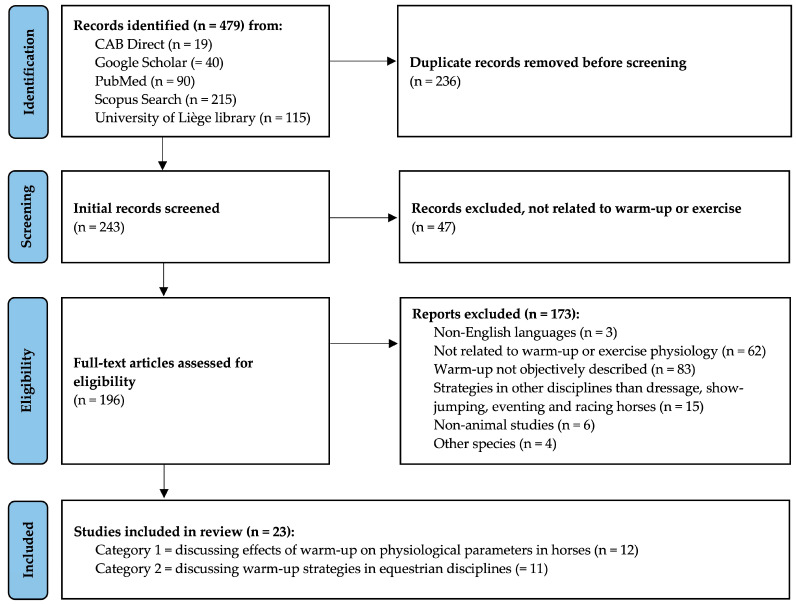
PRISMA flow diagram of the literature screening process, modified from [33].

**Table 1 animals-14-00945-t001:** Inclusion and exclusion criteria used to review the title and abstract of publications for appraisal of evidence on warm-up strategies and effects on performance in horses.

Inclusion Criteria	Exclusion Criteria
Original research articles or systematic reviews	Single case studies, personal opinions, non-peer reviewed studies, textbooks, or technical literature
Studies published in full and available in English	Studies only available as abstracts, or not available in English
Studies relating to equids including clinical case studies and trials, and in vivo equine models	Studies relating to other species than equids, or in vitro equine models
	Studies for which warm-up strategy was incompletely described and/or not objectively applied

**Table 2 animals-14-00945-t002:** Summary of the effects of warm-up (WU) on physiological parameters based on a systematic search of the published literature.

Study [Citation]	Study Design	Warm-Up and Studied Parameters	Main Results on the Effects of Warm-Up
Aerobic metabolism			
Lund, 1996 [34]	Design: randomized crossover, without versus with 2 different WU regimens Subjects: 6 TB Intervention: high-intensity exercise (105% VO_2max_) on treadmill	Warm-up regimens: Low-intensity: 5 min walk, 400 m canter, 5 min walk High-intensity: 5 min trot, canter until venous temperature > 39.5 °C, 5 min trot Parameters: VO_2_, HR, cardiac output, blood lactate	Low-intensity WU had beneficial effect on VO_2_ WU lowered peak plasma lactate concentration and its subsequent disappearance
Tyler, 1996 [35]	Design: randomized crossover, without versus with WU Subjects: 13 SB Intervention: high-intensity exercise (115% VO_2max_) on treadmill	Warm-up regimen: 5 min trot at 50% VO_2max_ Parameters: VO_2_, VCO_2_, total run time to fatigue, blood lactate	WU accelerated kinetics of gas exchange WU increased proportion of total energy requirement supplied by aerobic sources
McCutcheon, 1999 [36]	Design: randomized crossover, without versus with 2 different WU regimens Subjects: 6 SB Intervention: high-intensity exercise (115% VO_2max_) on treadmill	Warm-up regimens: Low-intensity: 10 min at 50% VO_2max_ High-intensity: 5 min at 50% VO_2max_ followed by 45 s intervals at 80, 90, and 100% VO_2max_ Parameters: VO_2_, total run time to fatigue, middle gluteal muscle biopsies, hematocrit, plasma total protein, blood lactate	WU was associated with higher aerobic energy contribution to total energy expenditure, lower glycogenolysis, and longer run time to fatigue WU decreased rate of blood and muscle lactate accumulation No additional benefit of high- versus low-intensity
Geor, 2000 [37]	Design: randomized crossover, without versus with 2 different WU regimens Subjects: 6 SB Intervention: high-intensity exercise (115% VO_2max_) on treadmill	Warm-up regimens: Low-intensity: 10 min at 50% VO_2max_ High-intensity: 7 min at 50% VO_2max_ followed by 45 s intervals at 80, 90, and 100% VO_2max_ Parameters: VO_2_, VCO_2_, CO_2_	WU accelerated VO_2_ and VCO_2_ kinetics WU decreased accumulated O_2_ deficit
Mukai, 2008 [38]	Design: randomized crossover, without versus with 2 different WU regimens Subjects: 11 TB Intervention: high-intensity exercise (115% VO_2max_) on treadmill	Warm-up regimens: Moderate-intensity: 1 min at 70% VO_2max_ High-intensity: 1 min at 115% VO_2max_ Parameters: VO_2_, VCO_2_, total run time to fatigue, blood lactate	WU increased VO_2_ peak values and decreased blood lactate accumulation during the first minute of intense exercise (suggesting greater aerobic than net anaerobic power) Higher time to fatigue following moderate-intensity WU
Mukai, 2010 [39]	Design: randomized crossover, with 3 different WU regimens Subjects: 9 TB Intervention: high-intensity exercise (115% VO_2max_) on treadmill	Warm-up regimens (canter): Low-intensity: 400 s at 30% VO_2max_ Moderate-intensity: 200 s at 60% VO_2max_ High-intensity: 120 s at 100% VO_2max_ Parameters: VO_2_, VCO_2_, CO_2_, HR, blood lactate	High-intensity WU accelerated VO_2_ kinetics High-intensity WU reduced reliance on net anaerobic power compared to low-intensity WU
Farinelli, 2021 [40]	Design: randomized crossover, without versus with WU Subjects: 6 MM horses Intervention: 50 min Marcha test	Warm-up regimen: 10 min walking at 10 km/h Parameters: HR, RR, blood lactate and glucose, CK, AST, serum cortisol	WU was not associated with changes in HR, RR, lactate, glucose, CK, AST, or cortisol directly after this predominantly aerobic intervention Faster HR recovery when horses performed WU
Thermoregulation			
Lund, 1996 [34]	Design: randomized crossover, without versus with 2 different WU regimens Subjects: 6 TB Intervention: high-intensity exercise (105% VO_2max_) on treadmill	Warm-up regimens: Low-intensity: 5 min walk, 400 m canter, 5 min walk High-intensity: 5 min trot, canter until venous temperature > 39.5 °C, 5 min trot Parameters: heat loss from airways, heat storage	Low-intensity WU had beneficial effect on heat balance (accumulation of heat was slower, despite higher body temperature at onset of maximal exercise) Low-intensity WU initiated sweating and promoted better thermoregulation
McCutcheon, 1999 [36]	Design: randomized crossover, without versus with 2 different WU regimens Subjects: 6 SB Intervention: high-intensity exercise (115% VO_2max_) on treadmill	Warm-up regimens: Low-intensity: 10 min at 50% VO_2max_ High-intensity: 5 min at 50% VO_2max_ followed by 45 s intervals at 80, 90, and 100% VO_2max_ Parameters: blood (right atrium) and middle gluteal muscle temperatures	WU increased muscle temperature No additional benefit from high- versus low-intensity
Geor, 2000 [37]	Design: randomized crossover, without versus with 2 different WU regimens Subjects: 6 SB Intervention: high-intensity exercise (115% VO_2max_) on treadmill	Warm-up regimens: Low-intensity: 10 min at 50% VO_2max_ High-intensity: 7 min at 50% VO_2max_ followed by 45 s intervals at 80, 90, and 100% VO_2max_ Parameters: blood temperature	Blood temperature increased after WU which was maintained throughout the exercise Increase in blood temperature depended on WU intensity: ○ Low-intensity: by 0.9 ± 0.1 °C after WU ○ High-intensity: by 1.9 ± 0.2 °C after WU
Mukai, 2008 [38]	Design: randomized crossover, without versus with 2 different WU regimens Subjects: 11 TB Intervention: high-intensity exercise (115% VO_2max_) on treadmill	Warm-up regimens: Moderate-intensity: 1 min at 70% VO_2max_ High-intensity: 1 min at 115% VO_2max_ Parameters: blood temperature (pulmonary artery)	WU exercise induced an increase in blood temperature, which was maintained throughout the whole sprint
Mukai, 2010 [39]	Design: randomized crossover, with 3 different WU regimens Subjects: 9 TB Intervention: high-intensity exercise (115% VO_2max_) on treadmill	Warm-up regimens (canter): Low-intensity: 400 s at 30% VO_2max_ Moderate-intensity: 200 s at 60% VO_2max_ High-intensity: 120 s at 100% VO_2max_ Parameters: blood temperature (pulmonary artery)	All WU regimens increased blood temperature Blood temperature during sprint was higher following high-intensity than low- and moderate-intensity WU
Buchner, 2017 [41]	Design: randomized crossover, without versus with 3 different WU regimens Subjects: 10 horses Intervention: examination before and after WU	Warm-up regimens: Regimen 1: 10 min whole-body vibration Regimen 2: 10 min extended walk Regimen 3: 8 min extended walk and 2 min trot Parameters: core and skin temperature, HR, RR	No difference in HR and core temperature after any WU regimens compared to no WU Slight increase in RR after walk and trot WU No difference in skin temperature after whole-body vibration Small increases in skin temperature after walk, and walk/trot WU
Janczarek, 2021 [42]	Design: randomized crossover, with 4 different WU regimens Subjects: 12 Warmblood horses Intervention: WU regimens of different durations in sand outdoor arena	Warm-up regimens: Very short: 10 min walk, 5 min trot, 5 min walk Short: 10 min walk, 10 min trot, 5 min walk Extended: 10 min walk, 15 min trot, 5 min walk Long-lasting: 10 min walk, 20 min trot, 5 min walk Parameters: body and mid-cannon surface temperature	WU increased body and surface temperatures, proportionally to its duration WU effect was achieved earlier and lasted longer in heavily trained horses than in non-performance horses
Farinelli, 2021 [40]	Design: randomized crossover, without versus with WU Subjects: 6 MM horses Intervention: 50 min Marcha test	Warm-up regimen: 10 min walking at 10 km/h Parameters: rectal temperature	WU increased rectal temperature before the Marcha test
Acid-base balance and biochemistry			
Frey, 1995 [43]	Design: randomized crossover, without versus after administration of sodium carbonate Subjects: 12 SB Intervention: race on track	Warm-up regimens: 2-mile slow or 1-mile fast Parameters: blood pH, HCO_3_^−^, PCO_2_, base excess, Na^+^, Ca^++^, Cl^−^, K^+^	Decreased PCO_2_, base excess and Ca^++^ after WU Increased K^+^ after WU
Fazio, 2012 [44]	Design: prospective observational Subjects: 10 healthy Italian saddle horses Intervention: WU and simulated show jumping competition	Warm-up regimen: 15 min (pacing, trotting, galloping, and 6 jumps 1.00–1.40 m) Parameters: Hematology and biochemical: lactate, bicarbonate, HCO_3_^−^, TCO_2_, O_2_ capacity and content, base excess of blood and extracellular fluid, pH, PCO_2_, PO_2_, SO_2_, hematocrit, and hemoglobin HR	Increased HR, lactate, TCO_2_, O_2_ capacity and content, base excess of blood and extracellular fluid, PCO_2_, PO_2_, SO_2_, hematocrit and hemoglobin after WU No difference in HCO_3_^−^ or pH after WU
Fazio, 2014 [45]	Design: prospective observational Subjects: 7 healthy Italian saddle horses Intervention: WU and simulated show jumping competition	Warm-up regimen: 15 min (pacing, trotting, galloping, and 6 jumps 1.20–1.40 m) Parameters: Serum: ALP, ALT, AST, CK, GGT, LDH, creatinine, urea, total bilirubin, glucose, Na^+^, Cl^−^, K^+^ HR and blood lactate	Increased HR, lactate, ALT, AST, CK, creatinine and K^+^ after WU Decreased glucose concentration after WU No difference in ALP, GGT, LDH, urea, total bilirubin, Na^+^ or Cl^−^ after WU

ALP: alkaline phosphatase; ALT: alanine transaminase; AST: aspartate transaminase; CK: creatine kinase; CO_2_: carbon dioxide; GGT: gamma-glutamyltransferase; HCO_3_^−^: bicarbonate; HR: heart rate; LDH: lactate dehydrogenase; MM: Mangalarga Marchador; O_2_: oxygen; PCO_2_: partial pressure of carbon dioxide; PO_2_: partial pressure of oxygen; RCT: randomized clinical trial; RR: respiratory rate; SB: Standardbred; SO_2_: oxygen saturation; TB: Thoroughbred; TCO_2_: total carbon dioxide; VO_2_: oxygen consumption or aerobic capacity; VO_2max_: maximal oxygen consumption or aerobic capacity; VCO_2_: rate of elimination of carbon dioxide.

**Table 3 animals-14-00945-t003:** Summary of the warm-up (WU) strategies in different disciplines based on a systematic search of the published literature.

Study [Citation]	Study Design	Warm-Up and Studied Parameters	Main Results on the Effects of Warm-Up
Racing horses			
Jansson, 2005 [46]	Design: randomized crossover, with 2 different WU regimensSubjects: 4 SB and 3 TB Intervention: SB: 2000 m trot TB: 800 m full speed	Warm-up regimens: short and long Parameters: rectal temperature, HR, RR, body weight	WU in SB Higher HR and RR 15 min post-exercise after long WU Increased temperature and body weight loss after long WU WU in TB No difference in HR, rectal temperature, or body weight loss Higher RR 5 to 15 min post-exercise after short WU
Dressage horses			
Murray, 2006 [47]	Design: observational Subjects: 267 competitors (104 Novice, 65 Medium, 60 Prix St-Georges, and 38 Grand Prix) Intervention: British Dressage tests	Warm-up regimen: free Parameters: time, final percentage score for each competition	Mean WU duration increased at higher levels Prix St-Georges and Grand Prix competitors spent more time at canter than Novice and Medium competitors Trot was main WU gait for Novice competitors, walk for others No effect of rider experience on WU strategy
Williams, 2009 [48]	Design: observational Subjects: 35 (16 Warmblood horses, 13 TB cross, and 6 TB) Intervention: 36 Elementary and 14 Medium levels of British Dressage tests	Warm-up regimen: free Parameters: video recordings, HR	No difference in mean and peak HR Positive correlation between mean HR during WU and competition WU duration between horses for different tests: ○ Different for Elementary tests ○ Not different for Medium tests
Chatel, 2021 [49]	Design: questionnaire Subjects: 132 European dressage riders Intervention: online survey (39 questions)		Reasons for performing WU Prepare the horse musculoskeletal system physically to work Get horses reactive to rider’s aids Increase suppleness WU strategies Walk reported as the main WU gait
Tranquille, 2021 [50]	Design: retrospective observational Subjects: 32 horses (12 elite [Intermediate I and above] and 20 non-elite [Medium and below]) ridden by 25 riders Intervention: British Dressage tests in field environment	Warm-up regimen: free up to 30 min Parameters: video recordings	Main WU gait was trot in both elite and non-elite horses No difference in WU duration between elite and non-elite horses Elite horses spent more time in canter than non-elite horses Non-elite horses spent more time in trot than elite horses No difference in time spent on left and right reins
Chatel, 2024 [51]	Design: retrospective observational Subjects: 39 WU sessions in 7 French horses (from Elementary up to Prix St-Georges levels) Intervention: flatwork sessions at home	Warm-up regimen: free Parameters: video recordings, post-WU form (within 12 h)	Main WU gait was walk WU sessions differed over time (range of 8 months) Riders accurately recalled 13.6% of WU routines No difference in WU duration or total time spent in walk during WU between air temperatures < 5 °C and >30 °C
Show jumping horses			
Whitaker, 2008 [52]	Design: observational Subjects: 87 competitors (49 Novice [0.90 m] and 38 Intermediate [1.20 m]) Intervention: British Show Jumping Association Show	Warm-up regimen: free Parameters: stop-watch	Mean WU duration and WU time spent walking were lower in Novice than Intermediate Total WU fences jumped and successfully jumped were lower in Novice than Intermediate
Tranquille, 2017 [53]	Design: observational Subjects: 10 elite horses ridden by 5 riders Intervention: World Class Performance 3-day training session	Warm-up regimen: free up to 30 min Parameters: video recordings, HR, inertial measurements units (linked to GPS)	Mean WU duration, time in each pace and on each rein did not differ over the 3 days Most horses spent more time in left canter Horses cantered slower, with a shorter stride length and longer stride duration during WU compared to course Mean, peak, and final WU HR did not change over the 3 days
Stachurska, 2018 [54]	Design: observational Subjects: 82 competitors Intervention: 1.20/1.30/1.35 m competitions	Warm-up regimen: free Parameters: video recordings, round scores	More jumps and higher obstacles during WU decrease performance WU duration did not affect the score Intensity of WU varied across the horses’ ages
Chatel, 2021 [49]	Design: questionnaire Subjects: 125 European show jumping riders Intervention: online survey (41 questions)		Reasons for performing WU Prepare the horse musculoskeletal system physically to work Get horses reactive to rider’s aids Decrease injury risk WU strategies Trot was reported as the main WU gait Most riders included 4–10 jumping efforts using different fence types
Chatel, 2021 [55]	Design: observational Subjects: 45 WU regimens across 27 riders and 29 horses Intervention: 1.30 m competitions	Warm-up regimen: free Parameters: video recordings	WU duration varied greatly among riders Walk was the main WU gait and trot the least prevalent used WU gait No difference between the number of faults in the show ring and WU routines
Chatel, 2024 [51]	Design: retrospective observational Subjects: 22 WU sessions in 3 horses (0.90 to 1.20 m level) Intervention: flatwork sessions at home	Warm-up regimen: free Parameters: video recordings, post-WU form (within 12 h)	Main WU gait was walk WU sessions differed over time (range of 8 months) Riders accurately recalled 13.6% of WU routines No difference in WU duration or total time spent in walk during WU between air temperatures < 5 °C and >30 °C
Eventing horses			
Valle, 2013 [56]	Design: observational Subjects: 10 Warmblood horses (5 at Intermediate and 5 at Advanced level) Intervention: two-day eventing competition	Warm-up regimen: free Parameters: HR, GPS (duration and speed), blood lactate	HR Higher mean and peak HR in Intermediate during WU of dressage and show jumping tests than Advanced No difference in HR between levels during WU of cross-country test WU duration Shorter WU in Intermediate for show jumping test than Advanced No difference in WU duration between levels during WU of dressage and cross-country tests

GPS: global positioning system; HR: heart rate; RR: respiratory rate; SB: Standardbred; TB: Thoroughbred.

**Table 4 animals-14-00945-t004:** Warm-up practices measured in observational studies in sport horses competing in Olympic disciplines. A description of the dressage levels is shown in Appendix B.

Discipline	Level (n)	Warm-Up Duration (Mean ± SD or SEM; min)	Mean Number of Jumps	Citation
Dressage	Novice (104)	25 ± 10		[47]
	Elementary (36)	31 ± 15		[48]
	Medium and below (20)	15 ± 6		[50]
	Medium (14)	31 ± 10		[48]
	Medium (65)	32 ± 12		[47]
	From Elementary to Prix St-Georges (39)	22 ± 6		[51]
	Prix St-Georges (60)	33 ± 11		[47]
	Inter I and above (12)	16 ± 6		[50]
	Grand Prix (38)	35 ± 10		[47]
Show jumping	0.90 m (49)	15 ± 1	10	[52]
	0.90–1.20 m (22)	27 ± 8		[51]
	1.10 m (38)	18 ± 1	13	[52]
	1.20/1.30/1.35 m (82)	25 ± 9	14	[54]
	1.30 m (45)	19 ± 12	9	[55]
	1.40 m (29)	18 ± 4	13	[53]
Eventing—Dressage	Intermediate (5)	38 ± 20		[56]
	Advanced (5)	35 ± 13		[56]
Show jumping	Intermediate—1.10 m (5)	16 ± 9		[56]
	Advanced—1.15/1.20 m (5)	32 ± 14		[56]
Cross-country	Intermediate—1.05 m (5)	28 ± 8		[56]
	Advanced—1.10/1.15 m (5)	28 ± 3		[56]

SD: standard deviation; SEM: standard error of the mean.

## Data Availability

No new data were created or analyzed in this study. Data sharing is not applicable to this article.

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
