# Peer review of "Warm-Up Strategies and Effects on Performance in Racing Horses and Sport Horses Competing in Olympic Disciplines"

_animals, 2024, doi:10.3390/ani14060945_

Round 1

Reviewer 1 Report

Comments and Suggestions for Authors

This manuscript includes a comprehensive collection of the equine literature on warm up; an area of interest for the athletic horse.  Overall the manuscript is clearly written and selects the major points from the described studies.   However there are areas where the manuscript could be improved to provide the reader with greater understanding of the area, so I hope that the suggestions below can be used to help improve the manuscript further.  Specifically:

The abstract and summary reasonably represent the manuscript

The introduction is a clear introduction to the subject area with clear aims. 

Results:

The tables are suitable and work in relation to the text.  However, in both the tables and the text it is not always clear what the differences are in the levels that were being assessed in the different studies, particularly for dressage.  For example Table 4. Elementary as been put at the first level listed, when in competition, the lowest level would be Novice, so it would seem to make more sense to put this first.  What is meant by intermediate in Table 4?  Does this mean FEI Intermediare level?  In which case this should be after Prix St George and listed with Inter 1.  Or does intermediate mean a level equivalent to medium or something else?  This is confusing for the reader.  These levels should be clarified from the original studies, and then rationalised in the text of this paper so that a reader can understand the comparison between the different previous studies.

Discussion

As the aim of this paper was to review the literature, is is important that this is a critical review of the literature and not just a repetition of the findings. The discussion section provides opportunity for the authors to critically evaluate the previous studies and discuss their findings in the light of what these mean - including comparing studies and providing suggestions for explanations of differences between studies or study findings.  At this time, the discussion is presented more as a repeat of the results and previous author conclusions with a small amount of discussion.  It would be significantly improved by reducing the amount of repetition of study findings and avoiding repeating author conclusions without discussion, but instead increasing discussion of what these results mean in the context of other studies, comparison with the literature in other species, and providing suggestions of the overall findings in the context of this overview.

The discussion would also benefit from a limitations section.

The conclusions do reflect the overall findings

Author Response

We thank the reviewer for the review, kind words and valuable suggestions. We enforced to address all comments. Please see the attachment for a point-to-point response to the reviewer.

Reviewer 2 Report

Comments and Suggestions for Authors

Much needed review of the evidence base on equestrian warm up research to date – I would like to also see (or at least for the authors to comment) the remit of warm up to also practice movements required for competition (well documented in the human field) within the document.

Simple summary

Line 10-11: Warm up (WU) should also prepare the horse for any movements required within competition

Concise summary would be useful to embed concept that equestrianism do not really engage in many different WU strategies e.g. passive vs dynamic as well as low vs high intensity or which meet individual horse needs.

Abstract

Good summary of studies reviewed. Again could consider passive vs dynamic WU in here

Keywords – be good to link in some equestrian disciplines maybe?

Introduction

Sets broad context and establishes rationale for study, personally would like to see more introduction of WU types / approaches maybe drawing on human strategies here (some provided in discussion but think across piece could be expanded) – could be tabulated for ease but would provide useful background for the reader

Materials and methods

Pity methods as reported do don’t fully align to systematic approach but still useful. Please clarify if only publication titles (and not wider) were reviewed to select potential papers.

Results

Nice presentation – like combination of summary tables with descriptive sections afterwards present an useful and informative overview

Discussion

Appropriate discussion  points raised; feel missed opportunity to introduce discussion of some of the aspects of equestrian practice which are not included in the papers (e.g. WU followed by waiting statically for competition; WU at home vs competition; rider and how their WU could impact horse) and scope to discuss future research opportunities in WU more and how these are essential to underpin accurate appraisal of horse welfare and links to SLO/ future of horse sports. As well as more discussion of lessons which horse sports could learn from human disciplines and emerging research. A note that warming down / cooling down also has little work conducted may be beneficial to add.

Conclusion

Appropriate

Author Response

(The authors gave the same response as above.)

Round 2

Reviewer 1 Report

Comments and Suggestions for Authors

The authors have considerably improved the manuscript and responded well to all suggestions.